# Comparison of Lateral Lumbar Interbody Fusion and Posterior Lumbar Interbody Fusion as Corrective Surgery for Patients with Adult Spinal Deformity—A Propensity Score Matching Analysis

**DOI:** 10.3390/jcm10204737

**Published:** 2021-10-15

**Authors:** Yu Matsukura, Toshitaka Yoshii, Shingo Morishita, Kenichiro Sakai, Takashi Hirai, Masato Yuasa, Hiroyuki Inose, Atsuyuki Kawabata, Kurando Utagawa, Jun Hashimoto, Masaki Tomori, Ichiro Torigoe, Tsuyoshi Yamada, Kazuo Kusano, Kazuyuki Otani, Satoshi Sumiya, Fujiki Numano, Kazuyuki Fukushima, Shoji Tomizawa, Satoru Egawa, Yoshiyasu Arai, Shigeo Shindo, Atsushi Okawa

**Affiliations:** 1Department of Orthopaedic Surgery, Graduate School of Medical and Dental Sciences, Tokyo Medical and Dental University, 1-5-45 Yushima, Bunkyo-ku, Tokyo 113-8510, Japan; matsukura.orth@tmd.ac.jp (Y.M.); morsorth@tmd.ac.jp (S.M.); hirai.orth@tmd.ac.jp (T.H.); yuasa.orth@tmd.ac.jp (M.Y.); inose.orth@tmd.ac.jp (H.I.); 060211ms@gmail.com (A.K.); utag.orth@tmd.ac.jp (K.U.); 0123456789jun@gmail.com (J.H.); egawa.orth@tmd.ac.jp (S.E.); okawa.orth@tmd.ac.jp (A.O.); 2Department of Orthopaedic Surgery, Saiseikai Kawaguchi General Hospital, 5-11-5 Nishikawaguchi, Kawaguchi 332-8558, Japan; kenitiro1122@gmail.com (K.S.); masaki0803.197571@gmail.com (M.T.); tori@zj8.so-net.ne.jp (I.T.); arai.orth@gmail.com (Y.A.); 3Department of Orthopaedic Surgery, Kudanzawa Hospital, 1-6-12 Kudanminami, Chiyoda-ku, Tokyo 102-0074, Japan; yamada.orth@tmd.ac.jp (T.Y.); kz_kusano@yahoo.co.jp (K.K.); ootani_k@kudanzaka.com (K.O.); shindo_s@kudanzaka.com (S.S.); 4Department of Orthopaedic Surgery, Yokohama City Minato Red Cross Hospital, 3-12-1 Shinyamashita, Naka-ku, Yokohama 231-8682, Japan; sumiya.orth.7077@gmail.com (S.S.); fnumano@js6.so-net.ne.jp (F.N.); 5Department of Orthopaedic Surgery, Saku General Hospital, 3400-28 Nakagomi, Saku 385-0051, Japan; kaz.fuku0628@gmail.com; 6Department of Orthopaedic Surgery, Tokyo Bay Urayasu Ichikawa Medical Center, 3-4-32 Toudaijima, Urayasu 279-0001, Japan; shoji.tomizawa@gmail.com

**Keywords:** adult spinal deformity (ASD), posterior lumbar interbody fusion, lateral lumbar interbody fusion, sagittal correction, perioperative complications, surgical invasiveness, degenerative adult deformity

## Abstract

Lateral lumbar interbody fusion (LLIF) is increasingly performed as corrective surgery for patients with adult spinal deformity (ASD). This paper compares the surgical results of LLIF and conventional posterior lumbar interbody fusion (PLIF)/transforaminal lumbar interbody fusion (TLIF) in ASD using a propensity score matching analysis. We retrospectively reviewed patients with ASD who received LLIF and PLIF/TLIF, and investigated patients’ backgrounds, radiographic parameters, and complications. The propensity scores were calculated from patients’ characteristics, including radiographic parameters and preoperative comorbidities, and one–to-one matching was performed. Propensity score matching produced 21 matched pairs of patients who underwent LLIF and PLIF/TLIF. All radiographic parameters significantly improved in both groups at the final follow-up compared with those of the preoperative period. The comparison between both groups demonstrated no significant difference in terms of postoperative pelvic tilt, lumbar lordosis (LL), or pelvic incidence–LL at the final follow-up. However, the sagittal vertical axis tended to be smaller in the LLIF at the final follow-up. Overall, perioperative and late complications were comparable in both procedures. However, LLIF procedures demonstrated significantly less intraoperative blood loss and a smaller incidence of postoperative epidural hematoma compared with PLIF/TLIF procedures in patients with ASD.

## 1. Introduction

Adult spinal deformity (ASD) is a disease defined as the deviation of the alignment of the spinal column that presents during adulthood. ASD is caused by a variety of conditions, such as de novo scoliosis, progressive adolescent idiopathic scoliosis, degenerative disc disease, iatrogenic kyphosis, and post-traumatic kyphosis [1,2,3]. Moreover, ASD causes a substantial and increasing burden on elderly patients and healthcare systems, as patients with ASD have a disability and a poor health-related quality of life (HR-QOL) [4,5]. Therefore, recognizing the importance of restoring sagittal balance in the surgical treatment of ASD has recently increased [3]. Thus, the goal of surgical treatment for patients with ASD is to achieve ideal sagittal alignment and balance, which are closely associated with pain and disability [6,7]. However, because of the high risk of perioperative complications despite the advances in surgical techniques and implant selection, surgical treatment for ASD remains challenging [8,9].

ASD has various surgical treatments (e.g., posterior interbody fusion (PLIF), transforaminal interbody fusion (TLIF), lateral lumbar interbody fusion (LLIF), and three-column osteotomy) [10,11]. A correction at the interbody space (e.g., LLIF and PLIF/TLIF) is a reasonable surgical method in patients with ASD and degenerative disc with kyphosis [1]. The current application of LLIF has increased in ASD patients [11,12,13]. Minimally invasive LLIF techniques are expected to reduce the risk of intraoperative bleeding and neurological damage, and thus LLIF may reduce the perioperative complications in the corrective surgery for ASD. Furthermore, a large interbody cage may have the potential to enhance spinal alignment correction in patients with ASD. However, few studies have investigated the surgical outcomes and risks for complications in comparison with LLIF and conventional PLIF/TLIF for patients with ASD [14].

Information regarding the surgical results of these treatment options is important in surgeons’ decision-making. Therefore, we investigated the results of radiographic parameters and surgical complications after LLIF and PLIF/TLIF for ASD patients in an elderly Japanese population. A propensity score matching analysis was conducted to minimize the selection bias of surgical procedures when comparing the surgical results of LLIF and PLIF/TLIF.

## 2. Materials and Methods

This multicenter study retrospectively reviewed ASD patients who were surgically treated between 1 January 2010 and 31 December 2016, at the hospital of this study and seven other affiliated hospitals. Institutional review board approval was obtained at each hospital for data collection. The inclusion criteria consisted of being ≥21 years of age at the time of surgery, a minimum follow-up period of 1 year with sufficient radiographic data, a surgery that included posterior instrumentation of ≥4 levels, and lower instrumented vertebra surgery of the pelvis with iliac screws or S2 ala-iliac screws in addition to S1 pedicle screws. The etiologies included degenerative kyphosis/kyphoscoliosis and post lumbar surgery. Patients with ASD caused by vertebral fractures were excluded.

Demographic data included age, sex, body mass index (BMI), medical comorbidities, location of the upper instrumented vertebra, number of intervertebral fusion levels, radiographic parameters, surgical procedure (i.e., LLIF, PLIF/TLIF, three-column osteotomy, or others), surgical invasiveness (intraoperative blood loss and surgical time), and incidence of surgical complications. Medical comorbidities (e.g., diabetes, renal dysfunction, cerebrovascular disease, cardiovascular disease, and respiratory disease) were registered. Radiographic parameters included the sagittal vertical axis (SVA), lumbar lordosis (LL), pelvic tilt (PT), pelvic incidence (PI), and spinopelvic harmony, which are evaluated by determining the PI minus LL (PI−LL) before surgery, 4 weeks after surgery, and at the final follow-up in standing position. Surgical complications are classified into perioperative and late complications. Perioperative complications are classified into surgery-related complications (e.g., epidural hematoma, postoperative neurological deficits, and surgical site infection) and systemic complications (e.g., cardiovascular events and deep vein thrombosis), which are usually observed soon after, or during, the operation. Late complications included proximal junction kyphosis (PJK), distal junction kyphosis, pseudarthrosis, rod breakage, or newly occurred vertebral fracture, which were generally caused by the stress on the implant or vertebra.

### 2.1. Surgical Procedures

This study investigated two surgical procedures, the LLIF and PLIF/TLIF. In the LLIF group in this study, first, multilevel LLIF was performed using the oblique lateral interbody fusion (OLIF, Medtronic, Minneapolis, MN, USA) or extreme lateral interbody fusion (XLIF, NuVasive Inc., San Diego, CA, USA) from L1–2 or L2–3 to L4–5, followed by posterior instrumentation. Schwab grade 1 or 2 osteotomies [15] were performed from L1–2 to L5–S1 using the posterior approach. L5–S1 PLIF/TLIF was then routinely performed using large lordotic cages, and lumbar lordosis was restored using a rod cantilever and compression technique. The instrumentation was performed from the lower thoracic spine/thoracolumbar junction to the sacrum/ilium. In the PLIF/TLIF group in this study, multilevel PLIF/TLIF combined with Schwab grade 1 or 2 osteotomies [15] were performed from L1–2 or L2–3 to L5–S1, and lumbar lordosis was restored using a rod cantilever and compression technique. The instrumentation was similarly performed from the lower thoracic spine/thoracolumbar junction to the sacrum/ilium with iliac screws or S2 ala-iliac screws, in addition to S1 pedicle screws. A hard brace was generally used for 3–6 months after surgery, regardless of the surgical procedures. The surgical procedure selection was determined at the discretion of each surgeon. Generally, surgeons were more likely to choose PLIF/TLIF when patients had histories of abdominal surgery or diseases, vascular abnormality, and difficulties in lateral access because of a high riding psoas muscle, high iliac crest, etc. LLIF tended to be chosen for patients who did not have the above factors, especially when surgeons wanted to avoid greater surgical invasiveness.

### 2.2. Statistics Analysis

Propensity score matching analysis was conducted to minimize the selection bias of surgical procedures by adjusting known confounding variables [16,17,18]. Furthermore, propensity scores for the surgical procedure (i.e., LLIF or PLIF/TLIF) were calculated with the following variables: the patient’s age and sex, BMI, medical comorbidities, number of intervertebral fusion levels, and radiographic parameters (SVA, LL, PI, and PI-LL at preoperation). The procedure was performed using a logistic regression model. The C-statistic suggested that the fitting was 0.77, which is a fairly good state. One-to-one matching of LLIF and PLIF/TLIF patients was performed based on propensity scores on the condition that the caliper was <0.4. After the matching, postoperative radiographic parameters, surgical invasiveness, perioperative complications, and late complications were compared between the two surgical procedures in the matched cases using a Mann–Whitney U test or chi-squared test. All statistical analyses were conducted using the Stata/MP version 14 (StataCorp, College Station, TX, USA), and *p*-values of <0.05 were considered as statistically significant.

## 3. Results

This study included 91 patients with full preoperative and postoperative radiographic data with a minimum 1 year follow-up. The cohort included 76 women and 15 men (mean age, 73.2 years; mean follow-up period, 24.2 months). Of the patients, 22 underwent LLIF and 69 underwent PLIF/TLIF. Propensity score matching resulted in 21 pairs of patients undergoing LLIF and PLIF/TLIF. Consequently, biases between the treatment groups diminished after the propensity score matching. The patients’ ages and sexes, BMIs, number of intervertebral fusion levels, and radiographic parameters before operation were adjusted (Table 1).

The LLIF group had significantly lower intraoperative blood loss (849 vs. 2359 mL). However, total surgical time was significantly longer in the LLIF group (536 vs. 421 min; Table 2). All parameters were significantly improved at 4 weeks after surgery and at the final follow-up compared with those at the preoperative period in both groups (Table 3). Consequently, most of the radiographic parameters were not significantly different between the two groups postoperatively. However, SVA tended to be smaller in the LLIF group at the final follow-up (Table 3).

Table 4 shows the incidence of surgical complications in both groups. No significant difference in the incidence of either overall local or systemic complications was observed in terms of perioperative complications. However, the incidence of epidural hematoma was significantly lower in the LLIF than in the PLIF/TLIF group (0.0% vs. 19.0%; *p* = 0.035). Moreover, the incidence of late complications was not significantly different between the two groups.

## 4. Discussion

The surgical invasiveness and the associated risks of complications in ASD surgeries remain problematic. Recently, the minimally invasive LLIF technique has been increasingly performed to potentially reduce the surgical risks of ASD surgeries [12,19,20,21]. However, insufficient information exists on whether LLIF indeed decreases the surgical invasiveness and the incidence of surgical compilations in ASD patients. Therefore, the use of PLIF/TLIF and the recently increased use of LLIF were compared, both of which are intervertebral corrections for the treatment of ASD. Furthermore, the propensity matching method was used to compare the surgical results in the two procedures. As surgical treatment for ASD is generally a high-risk surgery [8,9], conducting a randomized trial is difficult. The propensity score is a balancing score calculated by logistic regression analysis, which makes the distribution of measured baseline covariates similar between the two treatment groups [22]. In the current study, all of the covariates were successfully adjusted after one-to-one propensity score matching (Table 1), indicating no selection bias in the baseline characteristics between the LLIF and PLIF/TLIF groups.

Both the LLIF and PLIF/TLIF groups in this study resulted in favorable sagittal alignment correction, with significant improvements in SVA, LL, and PT, as well as PI−LL mismatch, compared with the preoperative parameters. In the comparison between the LLIF and PLIF/TLIF groups, SVA was smaller and the improvement in SVA from the preoperative value tended to be larger at the final follow-up in the LLIF group. However, we did not find marked differences between the two groups in terms of postoperative LL and PI−LL at 4 weeks after surgery and at the final follow-up. Previous studies have reported that LLIF is better able to correct sagittal imbalance than posterior corrective fusion in ASD patients. However, the surgical procedure of the posterior approach in these studies was the posterior spinal fusion with interbody fusion only at L5/S1, not multilevel PLIF/TLIF [11,23,24]. In this study, four-level interbody fusion (L2/3–5/S) was conducted in both the LLIF and PLIF/TLIF groups. In the LLIF procedure, interbody space can be lifted up using large cages, and may potentially have greater ability to restore segmental alignment. However, if the procedure is performed at multiple segments, the tightness of the anterior longitudinal ligament (ALL) limits the degree of lift-up and the correction of lumbar lordosis. Thus, the correction angle of LL in the LLIF group was slightly higher but not significant compared with that of the PLIF/TLIF group (LLIF, 44.1° ± 15.1°; PLIF/TLIF, 39.4° ± 15.5°). The release of ALL in addition to the LLIF procedure (anterior column realignment technique) would make a more radical correction [25,26], although the potential risk of vascular injury exists.

Since ASD mostly affects the elderly population, surgical invasiveness for ASD patients is a major problem [23,24]. Intraoperative blood loss is one of the major factors related to surgical invasiveness. Thus, this study demonstrated that the intraoperative blood loss of the LLIF group was reduced to one-third compared with the PLIF/TLIF group. In the PLIF/TLIF group, access to the intervertebral disc space was performed through the epidural space, and therefore, bleeding from the epidural venous plexus was inevitable at multilevel intervertebral discs. However, the LLIF group showed minimized epidural bleeding because access to the intervertebral disc space was performed through the retroperitoneal approach, except access to the L5/S level. Previous studies also reported the benefits of utilizing the lateral approach in reducing intraoperative blood loss [23,24]. Consequently, the total operation time in the LLIF group averaged 100 min longer than that in the PLIF/TLIF group. The PLIF/TLIF group was performed in this study through a single posterior approach, whereas the LLIF group was performed through the combined approach: the lateral oblique approach for anterior interbody fusion, and the subsequent posterior approach for pedicle screw fixation. Thus, the combined approach makes the total operating time much longer in the LLIF group. Previous reports have shown that blood loss is a risk factor for perioperative complications in lumbar fusion surgery; however, surgical time was not a significant factor [27,28,29]. Despite the longer operating time, LLIF procedures may have the benefit of reducing the bleeding and surgical risks through the combined approach. Additionally, two-stage surgery (performing anterior and posterior approaches on a separate day) can be selected for high-risk patients in LLIF procedures. Thus, the LLIF procedure is considered to be a good option for safe, corrective surgery for elderly ASD patients because patients with ASD are commonly older in age. Percutaneous posterior fixation, which is applicable for flexible patients, would further reduce surgical invasiveness [30,31,32,33,34].

The overall incidence of perioperative complications, including local and systemic complications, was not significantly different between the two groups in the present study. However, the incidence of epidural hematoma was significantly higher in the PLIF/TLIF group than in the LLIF group. The possible reasons for the reduced hematoma incidence are that the LLIF group in this study required minimal epidural manipulation, and that surgery-related bleeding occurred significantly less in the LLIF group compared to the PLIF/TLIF group. Previous studies reported that the abundant intraoperative blood loss was a significant risk factor for early perioperative complications in the fusion and instrumentation of degenerative lumbar scoliosis [27,28,29]. Therefore, the LLIF procedure may potentially reduce surgical risks for ASD patients. In terms of late complications, no differences were noted in overall incidence, implant failure, PJK, or newly occurred vertebral fractures between the two groups. This result is similar to previous reports of comparisons between LLIF and single posterior approaches in surgical treatments for ASD [11,23,24,35]. We generally use hard braces after the surgery for 3–6 months. However, the long-term use of hard braces is reported to cause complications in elderly patients [36]. It was also reported that bracing after deformity-correction surgery did not effectively reduce PJK [37]. At the final stages, shifting from a hard brace, to a soft brace, or a dynamic brace [38] may be important for avoiding brace-related complications, preserving back muscle strength and improving QOL.

This study has several limitations. The main limitations are the small number of patients and their short-term follow-up. Further studies should involve a larger sample size and a longer follow-up period. In addition, HR-QOL was not investigated in this study. Previous studies have reported that lumbar spinal fusions and corrective surgeries are effective in improving QOL [6,24,39,40,41]. Further studies should be conducted using HR-QOL evaluations, such as the Oswestry Disability Index, SRS-22, and SF-36. Lastly, the choice of surgical approaches was not randomized in this study. A propensity score matching analysis was conducted to minimize the selection bias of the surgical procedures.

Despite these limitations, this is the first study that compared multilevel LLIF and PLIF/TLIF procedures for ASD patients using propensity score matching methods, clearly showing that intraoperative bleeding and postoperative hematoma were reduced in the LLIF procedures.

## 5. Conclusions

In comparison with LLIF and PLIF/TLIF procedures for the correction of ASD, the majority of the radiographic parameters were not significantly different between the two groups postoperatively. However, SVA tended to be smaller in the LLIF group at the final follow-up. Overall perioperative complications were comparable in LLIF and PLIF/TLIF for patients with ASD. However, LLIF procedures demonstrated significantly less intraoperative blood loss and a smaller incidence of postoperative epidural hematoma.

## Figures and Tables

**Table 1 jcm-10-04737-t001:** Patient characteristics in the lateral lumbar interbody fusion (LLIF) group and the posterior lumbar interbody fusion (PLIF)/transforaminal lumbar interbody fusion (TLIF) group after the matching.

Parameter	LLIF (N = 21)	PLIF/TLIF (N = 21)	*p*-Value
Age at surgery (years)	74.0 ± 7.6	73.2 ± 7.3	0.687
Sex (male/female: cases)	2/19	2/19	1.000
BMI	21.7 ± 4.0	22.1 ± 3.3	0.823
Medical comorbidity (yes/no)	8/13	5/16	0.317
SVA (mm)	153.7 ± 78.9	148.3 ± 48.6	0.763
LL (deg.)	1.1 ± 12.6	1.6 ± 12.1	0.930
PI (deg.)	51.6 ± 9.1	50.0 ± 9.1	0.496
PI-LL (deg.)	50.5 ± 14.5	48.4 ± 11.9	0.660
PT (deg.)	37.1 ± 15.4	32.1 ± 7.7	0.162
TK (T4-12) (deg.)	28.4 ± 16.7	20.9 ± 16.5	0.290
No. of fixed levels	8.4 ± 1.9	7.7 ± 1.52	0.548
Type of anchor for pelvis (iliac screw/S2-ala-iliac screw: cases)	15/6	19/2	0.116

Mean ± standard deviation; LLIF, lateral lumbar interbody fusion; PLIF, posterior lumbar interbody fusion; TLIF, transforaminal lumbar interbody fusion; BMI, body mass index; SVA, sagittal vertical axis; LL, lumbar lordosis; PI, pelvic incidence; PT, pelvic tilt; TK, thoracic kyphosis; deg., degree; No., number.

**Table 2 jcm-10-04737-t002:** Surgical invasion in the LLIF group and the PLIF/TLIF group.

Parameter	LLIF (N = 21)	PLIF/TLIF (N = 21)	*p*-Value
Surgical time (min)	535.9 ± 123.1	426.8 ± 96.2	<0.001 *
Estimated blood loss (grams)	848.7 ± 477.1	2358.6 ± 1911.6	<0.001 *

Mean LLIF, lateral lumbar interbody fusion; PLIF, posterior lumbar interbody fusion; TLIF, transforaminal lumbar interbody fusion; ± standard deviation; LLIF, lateral lumbar interbody fusion; PLIF, posterior lumbar interbody fusion; TLIF, transforaminal lumbar inter-body fusion; *, *p* < 0.05.

**Table 3 jcm-10-04737-t003:** The comparison of the LLIF group and the PLIF/TLIF group for radiographic parameters at each follow-up time.

Parameter	LLIF (N = 21)	PLIF/TLIF (N = 21)	*p*-Value
At 4 weeks after surgery			
SVA (mm)	24.1 ± 41.7	33.8 ± 41.4	0.279
ΔSVA(mm)	−129.6 ± 76.9	−114.5 ± 51.6	0.725
LL (deg.)	45.2 ± 7.8	41.0 ± 10.9	0.268
ΔLL (deg.)	44.1 ± 15.1	39.4 ± 15.5	0.473
PI-LL (deg.)	6.4 ± 8.9	9.1 ± 13.9	0.920
ΔPI-LL (deg.)	−44.1 ± 15.1	−39.0 ± 15.5	0.473
PT (deg.)	21.5 ± 15.4	21.4 ± 6.1	0.890
ΔPT (deg.)	−15.6 ± 9.2	−11.0 ± 5.4	0.057
TK (T4-12) (deg.)	36.7 ± 11.8	32.3 ± 13.2	0.473
ΔTK (T4-12) (deg.)	8.3 ± 13.3	11.4 ± 13.1	0.562
At final follow-up			
SVA (mm)	23.2 ± 37.6	52.4 ± 41.4	0.044 *
ΔSVA (mm)	−130.4 ± 84.2	−95.9 ± 45.0	0.097
LL (deg.)	43.9 ± 9.4	40.1 ± 10.1	0.420
ΔLL (deg.)	42.8 ± 15.6	38.5 ± 18.3	0.513
PI-LL (deg.)	7.8 ± 9.4	9.9 ± 16.3	0.753
PT (deg.)	25.1 ± 14.7	21.6 ± 7.5	0.092
ΔPT (deg.)	−12.0 ± 12.3	−11.0 ± 6.9	0.705
TK (T4-12) (deg.)	42.8 ± 15.6	36.7 ± 14.9	0.273
ΔTK (T4-12) (deg.)	16.7 ± 13.1	15.8 ± 15.1	0.830
Loss of correction from 4 weeks postoperatively at last observation			
SVA (mm)	−0.1 ± 49.1	18.6 ± 31.0	0.326
LL (deg.)	1.3 ± 3.1	0.9 ± 4.0	0.638
PT (deg.)	3.6 ± 11.1	0.3 ± 5.9	0.289
TK (T4-12) (deg.)	6.0 ± 6.7	4.4 ± 7.7	0.434

Mean ± standard deviation; LLIF, lateral lumbar interbody fusion; PLIF, posterior lumbar interbody fusion; TLIF, transforaminal lumbar inter-body fusion; SVA, sagittal vertical axis; LL, lumbar lordosis; PI, pelvic incidence; PT, pelvic tilt; TK, thoracic kyphosis; deg., degrees; **, p* < 0.05.

**Table 4 jcm-10-04737-t004:** Surgical complications of the LLIF group and the PLIF/TLIF group.

Parameter	LLIF (N = 21)	PLIF/TLIF (N = 21)	*p*-Value
Perioperative complications			
Local complications (yes/no)	4/17 (19.0%)	6/15 (28.6%)	0.454
Neurological deficit (yes/no)	4/17 (19.0%)	3/18 (14.3%)	0.679
Epidural hematoma (yes/no)	0/21 (0.0%)	4/17 (19.0%)	0.035 *
Surgical site infection (yes/no)	1/20 (4.8%)	0/21 (0.0%)	0.261
Systemic complications (yes/no)	1/20 (4.8%)	1/20 (4.8%)	1.000
	Cerebrovascular events: 1	Deep vein thrombosis: 1	
Total (yes/no)	5/16 (23.8%)	7/14 (33.3%)	0.482
Late complications			
Implant failure (yes/no)	1/20 (4.8%)	1/20 (4.8%)	1.000
Proximal junctional kyphosis (yes/no)	5/16 (31.3%)	5/16 (31.3%)	1.000
Newly occurred vertebral fracture (yes/no)	3/18 (14.3%)	3/18 (14.3%)	1.000
Total (yes/no)	7/14 (33.3%)	6/15 (28.6%)	0.739
Revision surgery	2/19 (9.5%)	1/20 (4.8%)	0.549

Mean *: *p* < 0.05.

## Data Availability

The datasets generated during and/or analyzed during the current study are available from the corresponding author on reasonable request.

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
