# Peer review of "Comparison of Lateral Lumbar Interbody Fusion and Posterior Lumbar Interbody Fusion as Corrective Surgery for Patients with Adult Spinal Deformity—A Propensity Score Matching Analysis"

_jcm, 2021, doi:10.3390/jcm10204737_

Round 1
Reviewer 1 Report
Due to the fact that no significant difference can be found, the different views/arguments in the choice of surgical access are subject to individual experience. This recommendation could be expressed a little more clearly or, if necessary, the decisive criteria why the respective access was chosen could be a little more precisely. Thus, the reader could also be provided with a recommendation for the appropriate access to be selected.
Unfortunately, both cohorts include a dorsal access L5/S1. This segment must be considered a "key segment" in the correction of lumbar lordosis. Also, late complications in the sense of pseudarthrosis or screw loosening occur especially in this segment with long instrumentations. Here, the use of an anterior approach with all the described advantages of the larger implant could possibly make a significant difference. Since this cannot be changed retrospectively, it would at least be of interest to know how what ,mind of instrumentation into the sacrum was used (S1, S2, S2 ala-ilium or ilium). This and also the choice of additive intercoporeal support has been shown to have an influence on stability and the occurrence of complications with regard to the L5/S1 segment.
Despite certain weaknesses in the selection of the cohort to be studied, this paper is worthy of publication after appropriate supplementation.
Reviewer 2 Report
Dear Authors
I have reviewed your paper with great interest.
My revision is:
Title: Very Good
Abstract: Very Good
Introduction and AIM: The problem and the aim are well descripting.
In elderly spine deformities you can use only a dynamic cast, please cite and discuss this paper:
Meccariello L, Muzii VF, Falzarano G, Medici A, Carta S, Fortina M, Ferrata P. Dynamic corset versus three-point brace in the treatment of osteoporotic compression fractures of the thoracic and lumbar spine: a prospective, comparative study. Aging Clin Exp Res. 2017 Jun;29(3):443-449. doi: 10.1007/s40520-016-0602-x. Epub 2016 Jul 8. PMID: 27386868.
Marterials, Patients and methods and statistics: All good.
Results: Focus on and well described.
Discussion and Thread: effectiveness Focus ON.
Cite and Discuss your clinical results with this paper:
Cervera Irimia J, Tomé-Bermejo F, Piñera-Parrilla AR, Benito Gallo M, Bisaccia M, Fernández-González M, Villar-Pérez J, Fernández-Carreira JM, Orovio de Elizaga J, Areta-Jiménez FJ, Álvarez Galovich L, Rollo G, Caruso L, Meccariello L. Spinal fusion achieves similar two-year improvement in HRQoL as total hip and total knee replacement. A prospective, multicentric and observational study. SICOT J. 2019;5:26. doi: 10.1051/sicotj/2019027. Epub 2019 Jul 30. PMID: 31359861; PMCID: PMC6664676.
Posteriori fusion is always the gold standard?
Please cite discuss these papers:
Medici A, Meccariello L, Falzarano G. Non-operative vs. percutaneous stabilization in Magerl's A1 or A2 thoracolumbar spine fracture in adults: is it really advantageous for a good alignment of the spine? Preliminary data from a prospective study. Eur Spine J. 2014 Oct;23 Suppl 6:677-83. doi: 10.1007/s00586-014-3557-7. Epub 2014 Sep 12. PMID: 25212447.
Cervera-Irimia J, González-Miranda Á, Riquelme-García Ó, Burgos-Flores J, Barrios-Pitarque C, García-Barreno P, García-Martín A, Hevia-Sierra E, Rollo G, Meccariello L, Caruso L, Bisaccia M. Scoliosis induced by costotransversectomy in minipigs model. Med Glas (Zenica). 2019 Aug 1;16(2). doi: 10.17392/1015-19. Epub ahead of print. PMID: 31223011.
Short segment fixation of thoracolumbar fractures with pedicle fixation at the level of the fracture
Caruso, L., Bisaccia, M., Rinonapoli, G., ...Filipponi, M., Rollo, G. this link is disabled,
References: Well chosen but to improve
Figures and Table: Very Good.
Round 2
Reviewer 2 Report
Dear Author
Well Done I accept your paper!